# Community-for-Care: An Integrated Response to Informal Post-Caregivers

**DOI:** 10.3390/healthcare13243318

**Published:** 2025-12-18

**Authors:** Catarina Inês Costa Afonso, Ana Spínola Madeira, Alcinda Reis, Susana Magalhães

**Affiliations:** 1RISE-Health, Santarém School of Health, Polytechnic Institute of Santarém, Quinta do Mergulhão Srª da Guia, 2005-075 Santarém, Portugal; 2i3S—Instituto de Investigação e Inovação em Saúde, Universidade do Porto, 4200-135 Porto, Portugal

**Keywords:** informal post-caregivers, support network, compassionate communities, care literacy, coordinated care, bereavement support, community health

## Abstract

Background/Objectives: Informal caregivers play a crucial role in healthcare, but when caregiving ends the “post-caregivers” often remain invisible and unsupported. Post-caregivers face needs such as reconstructing their identity and finding space and time to grieve. This study aimed to design a support network for informal post-caregivers by exploring perceptions of diverse stakeholders. Methods: A qualitative inductive study was conducted using three focus groups (*n* = 15; ages 35–70; 12 women, 3 men) held online between June and July 2023. Participants included palliative care team members, home support professionals, general practitioners, informal caregivers, post-caregivers, and members of civil society. A semi-structured guide was used, and narratives were analyzed with a Narrative Medicine-informed approach and thematic analysis. Results: Community-For-Care emerged as an overarching and distinctive concept that, while aligned with the ethos of Compassionate Communities, specifically addresses the transition after caregiving ends, a phase largely absent from existing models. It symbolizes the “living forces of the community” mobilized to accompany informal post-caregivers through identity reconstruction, bereavement, and reintegration. Three interrelated thematic axes structure this concept: (1) Compassion Axis—emphasizing a compassionate community that values caregiving; (2) Coordinated Action Axis—highlighting coordinated, continuous support across healthcare and community services; and (3) Care Literacy Axis—underscoring education and training for caregivers, post-caregivers, and professionals. These axes dynamically interact to empower post-caregivers and stitch the holes in the support network. Conclusions: A community-centered, post-caregiver-focused framework such as Community-For-Care offers a novel extension of compassionate communities by directly addressing the loneliness, identity rupture, and invisibility that often characterize the transition after caregiving. Reinforcing compassion, coordinated action, and care literacy can enable communities to better acknowledge the contributions and ongoing needs of post-caregivers, supporting their emotional recovery, social reintegration, and reconstruction of daily life. By integrating these three axes into community practice, the model introduces a post-care-specific structure that can enhance well-being, reduce preventable health decline, and relieve pressure on formal services by mobilizing local, civic, and relational assets.

## 1. Introduction

Informal caregivers who cease providing care, whether due to hospitalization or the death of the care recipient, become post-caregivers [1]. These individuals frequently report a profound sense of emptiness and loss of purpose, struggling to rebuild daily routines and identities that were once defined by their caregiving role [2,3,4,5]. Concurrently, they face bereavement and must undertake significant emotional, social, and existential adjustments [3,4,5]. The needs of informal caregivers and post-caregivers are often overlooked in research and policy, despite evidence that informal caregiving contributes substantially to health care provision and costs at both individual and system levels [6,7,8]. While caregiving can be rewarding and meaningful [9], it is also associated with considerable financial, physical, and psychological strain [10,11,12].

Informal caregivers provide essential care that sustains health systems, yet they often do so at personal cost—reducing work hours, leaving employment, and risking long-term financial insecurity [13]. Gender, socioeconomic status, ethnicity, and care intensity interact to shape caregiving experiences and outcomes, highlighting the role of intersectionality in influencing well-being both during and after caregiving [14,15,16]. Women continue to assume a disproportionate share of caregiving responsibilities [14,15], and caregivers from disadvantaged socioeconomic or ethnic backgrounds frequently lack access to supportive services, exacerbating health and social inequalities [14,15,16,17]. As a result, caregiving is not a uniform experience but one mediated by structural and cultural determinants that have implications for post-care health and resilience.

The transition out of caregiving represents a complex psychosocial process. Although some post-caregivers experience relief and improved well-being, many encounter prolonged grief, depression, and social isolation that can persist for months or years [17,18,19,20]. Larkin [18] delineated the post-caregiving journey in three phases: “emptiness in aftercare” (immediate feelings of loss and purposelessness), “closing the time of caring” (adapting routines and settling affairs in the months after caregiving), and “rebuilding life after caring” (gradually redefining oneself and re-engaging with life). Bereavement, however, remains a critical and often unaddressed dimension of this transition. Post-caregivers frequently become “hidden patients”, presenting with fatigue, anxiety, and unresolved grief, which are seldom recognized within formal health systems [2,3,4].

Other studies reinforce the idea that former caregivers undergo a significant transition that benefits from support [20,21,22]. A recent qualitative study [22] using grounded theory likewise found that the transition for post-family caregivers extends over years and involves critical moments they termed: “post-caring emptiness”, “the end of the caregiver period”, and “movement towards a new life”. They observed that this transition often begins even before the caregiving ends (as the caregiver anticipates the loss) and can continue for more than three years afterward. Notably, family and professional support were found to be crucial during these phases, and post-caregivers themselves can become a source of support for others in similar situations. These findings align closely with Larkin’s phases and highlight that structured interventions could help post-caregivers navigate the emotional journey and practical adjustments in their post-care lives.

Empirical studies confirm that post-caregivers display elevated rates of depression, anxiety, and physical morbidity extending well beyond the caregiving period [3,4,20,21]. Persistent physiological stress responses, such as disrupted sleep and dysregulated cortisol levels, may contribute to long-term health deterioration [3]. Yet, despite the scale and persistence of these challenges, formal support for post-caregivers remains scarce. Health and social care systems largely focus on patients and active caregivers, with minimal attention to those navigating life after care ends [4,5,23]. This gap, namely, the absence of structured, community-based models specifically designed to address the post-caregiving transition, constitutes a critical unmet need in the literature and in practice.

In Portugal, the Statute of the Informal Caregiver (Law No. 100/2019) represents a significant policy milestone, granting rights to training, temporary substitution, and financial assistance. However, its implementation has been uneven and geographically inconsistent, leaving many caregivers without effective access to formal support [24]. According to the European Commission, informal caregiving in Portugal accounts for approximately €4 billion annually, around 2% of the national GDP, reflecting its economic and social importance [25]. Nonetheless, many caregivers, particularly women aged 55–70, face barriers to re-entering the workforce after caregiving due to skill depreciation, health decline, and age-related employment discrimination [26]. This structural vulnerability is compounded by limited pension entitlements, contributing to the heightened risk of poverty among older post-caregivers.

Emerging conceptual frameworks, including Narrative Medicine and Compassionate Communities, offer innovative approaches to addressing the unmet needs of post-caregivers. Narrative Medicine, rooted in the work of Charon [27] and further elaborated by Palla et al. [28], emphasizes narrative competence, the ability to acknowledge, absorb, and act upon the stories of illness and care, to foster empathy, ethical awareness, and person-centered care. Through storytelling and reflective writing, post-caregivers can articulate experiences of loss and recovery, providing insights into the psychosocial dimensions of the post-care phase.

The Compassionate Communities model, derived from the work of Abel and Kellehear [29], extends public health palliative care principles by mobilizing civic and community networks to share responsibility for caring, dying, death, and bereavement. Within this paradigm, care is viewed as a collective, community-embedded endeavor rather than an individual or institutional task. Compassionate Communities create supportive ecosystems that engage family, neighbors, and local organizations in sustaining people through illness and bereavement [29,30]. However, while Compassionate Communities address bereavement broadly, they do not explicitly focus on the specific identity, social, and health transitions of post-caregivers. This creates a conceptual gap that the present study seeks to address.

The study explores narratives from a range of stakeholders to imagine a supportive network for post-caregivers. Focus groups were specifically used to generate a multi-perspective dialogue, and narratives were analyzed through thematic analysis with sensitivity to narrative elements (a hybrid approach marrying Narrative Medicine and qualitative thematic coding). This design aimed to surface not only individual experiences but also collective insights and areas of consensus regarding community action. To strengthen methodological alignment, the study was guided by the following research questions: What needs and gaps in support do stakeholders identify for informal post-caregivers? What formal and informal community resources could be mobilized to support individuals after caregiving ends? How do stakeholders envision community-based strategies or structures that could sustain post-caregiver well-being? These questions structured the focus group discussions and underpin the findings presented in the Section 3, which coalesce around three thematic axes—compassion, coordinated action, and care literacy—that together form the basis of the Community-For-Care framework, a novel extension of existing models that specifically centers the post-caregiver experience.

## 2. Methods

### 2.1. Study Design

This was a qualitative study with an inductive, exploratory design, utilizing focus group discussions as the primary data collection method. The study was informed by principles of Narrative Medicine and employed thematic analysis as described by Braun and Clarke [31]. Narrative Medicine, as conceptualized by Charon [27], emphasizes the importance of stories and personal narratives in healthcare, advocating for close listening and reflection to better understand patients’ (or caregivers’) experiences. In this study, Narrative Medicine provided a guiding ethos: the researchers treated each participant’s shared story and perspective as valuable narrative data, paying attention to the language, emotions, and meaning within their accounts. This narrative sensibility was applied during both data collection (by encouraging storytelling and reflection in the focus groups) and data analysis (through “close reading” of transcripts and iterative interpretation by the researchers).

The methodological framework was qualitative and inductive, meaning the researchers did not impose a predetermined model on the data but rather let themes emerge from participants’ discussions. We chose focus groups to capitalize on interactive dynamics; the group setting can generate richer descriptions and insights as participants react to and build upon each other’s. Given our interest in community and network concepts, the focus group format mirrored a communal discussion, which was appropriate for brainstorming a community-centered intervention. Recommended practices for focus groups in applied research [32] were followed, which include careful moderation, a structured question route, and attention to group dynamics.

### 2.2. Participants and Setting

Participants were purposively sampled to include a diverse range of stakeholders in the caregiving continuum. The inclusion criteria were: Age 18 or older; ability to understand and speak Portuguese; willingness to discuss the topic; and belonging to one of the following categories—health professional in palliative care or home care, general practitioner/family doctor, member of a caregiver support association or community support institution, current informal caregiver, or post-informal caregiver (post-caregiver). The goal was to capture different perspectives (professional, personal, organizational, lay public) regarding support for post-caregivers. Individuals who did not meet these criteria or who were unable to participate in an online group discussion were excluded.

The researchers collaborated with various official bodies and networks to recruit participants: caregiver associations, primary care and palliative care teams, community centers, and advocacy groups for caregivers. These organizations disseminated our study invitation to their members. Interested individuals then contacted the research team and were given more detailed information. We used an online registration form to collect basic information and schedule participants. This network-based recruitment ensured credibility and trust, as participants were reached through organizations they were already affiliated with. Although, participants might be influenced by a greater awareness of caregiver advocacy issues. Their perspective, particularly in emphasizing community responsibilities and systemic gaps, was essential for capturing firsthand experiential knowledge and community-level insights, which were foundational to the study’s aims.

Participants were organized into three focus groups, largely grouping similar backgrounds together to encourage open sharing. The rationale was that some participants (e.g., post-caregivers) might speak more freely about sensitive experiences when not in the same room as healthcare providers who may have been involved in their care. Thus, the composition was as follows:Focus Group 1: Health professionals—including members of palliative care teams, home support nurses, and general practitioners/family physicians.Focus Group 2: Members of civil society and representatives of community caregiver support organizations (e.g., leaders or volunteers in caregiver associations, social workers, community activists).Focus Group 3: Current informal caregivers and informal post-caregivers (post-caregivers).

Each focus group was intentionally heterogeneous within a shared context, meaning participants had broadly similar roles in caregiving but came from different regions or institutions. This diversity aimed to generate a wide range of experiences while the common ground (role) facilitated understanding among participants. A total of 15 individuals participated (5 in FG1, 5 in FG2, 5 in FG3). Ages ranged from 35 to 70 years, and there were 12 women and 3 men, reflecting the fact that caregiving roles (both formal and informal) are often filled by women. Participants included: two representatives of caregiver support organizations, one nursing researcher, one family physician, two hospital-based internal medicine physicians with palliative care expertise, two psychologists (one in palliative care, one in mental health), four representatives of caregiver associations (some of whom were also post-caregivers themselves), one current family caregiver, and two informal post-caregivers (one who had cared for a spouse, one for a parent). A balanced representation of firsthand caregiving experience and professional insight was achieved. The number of participants is consistent with methodological recommendations for exploratory qualitative research using focus groups, where saturation is typically reached within two to three groups when participants share a common experiential domain. After the second group, no new codes appeared, and the third group confirmed thematic consistency, indicating conceptual saturation. Moreover, the heterogeneity within each role-based group (participants from different institutions, regions, and caregiving contexts) provided sufficient variability to ensure depth while still enabling convergence of themes relevant to the research questions.

### 2.3. Data Collection

All focus groups were conducted via a secure online video conferencing platform, as this allowed us to include participants from different regions of Portugal without requiring travel. The online format was also convenient given the ongoing need for caution with group meetings (the study took place in 2023, in the context of post-COVID adjustments). Each session lasted approximately 60 min. A trained moderator (one of the researchers, CA) facilitated the discussions, and a co-moderator (SM) acted as an observer and note-taker. The moderator and observer introduced themselves at the start, explaining their roles and interest in the topic, and made efforts to create a friendly, respectful atmosphere. Participants were reminded of the purpose of the study and the importance of all voices being heard.

A semi-structured interview guide with open-ended questions was used to steer the discussion (see Appendix A). The guide was developed based on literature and a pilot group, and it focused on exploring participants’ perceptions of the needs of post-caregivers and ideas for a support network. Key questions included, for example: “When you hear the word ‘caregiving’, what comes to mind?” (to warm up and establish context); “What do you hear people saying about resources to support informal post-caregivers?”; “On a scale of 1 to 5, how concerned are you about this topic, and why?”; “How widespread do you think a support network for post-caregivers is (or should be)?”; “Should action be taken, and if so by whom?”; “What might you personally do about this issue?”; and “What advice do you have for community leaders regarding this problem?”. These questions progressed from general impressions to specific calls for action, in line with Krueger & Casey’s framework [32] for issue-focused focus groups. The inclusion of a scaled question (1 to 5 concern level) was a technique to prompt reflection and comparative discussion; while numeric responses are unusual in Narrative Medicine, here it served as a catalyst for storytelling (participants explained the reasons behind their concern rating, often sharing personal anecdotes).

Throughout the focus groups, the moderator employed Narrative Medicine techniques to deepen the discussion: for instance, asking participants to elaborate on particular phrases they used (“You mentioned ‘lonely’—could you tell us more about what that loneliness looks like?”) and occasionally posing reflective prompts (“Imagine a community where post-caregivers feel fully supported—what would that look like?”). The co-moderator took detailed field notes on group dynamics, noting who spoke and when, moments of agreement or tension, emotional tone, and non-verbal cues (as visible on video). These observations helped later in interpreting how the group context may have influenced the content (for example, whether any voices were dominant or if consensus emerged spontaneously).

Each session concluded with a reflective summary by the moderator, designed to acknowledge participants’ contributions and elicit any final clarifications or additions. To safeguard emotional well-being, particularly given the sensitive nature of post-caregiving experiences, the moderator was equipped to pause discussions and provide referrals to counseling resources if necessary. Although several participants expressed visible emotional responses when recounting bereavement or caregiving-related distress, all opted to continue after brief acknowledgments of their experiences. To minimize potential social desirability bias, especially in sessions involving heterogeneous stakeholder groups, participants were stratified by role, and the moderator emphasized the value of critical and honest perspectives. Field notes indicated active participation from all group members, with no individual or subgroup dominating the dialogue. These observations suggest that power asymmetries were appropriately moderated.

All focus group discussions were audio-recorded (with permission) for accuracy and later transcribed verbatim. Any identifying information mentioned (such as specific names of people or places) was redacted in transcripts to ensure confidentiality.

Each focus group adhered to this structure but allowed flexibility for conversation flow. The moderator ensured all questions were touched upon, while also following relevant tangents if they arose (for example, if participants started discussing a specific local initiative, the moderator let that conversation proceed as it yielded concrete ideas).

### 2.4. Data Analysis

Data analysis was performed using thematic analysis in an inductive manner. After transcription, both researchers (CA and SM) independently read through each focus group transcript multiple times to immerse themselves in the data and obtain a holistic sense of each discussion. Initially, open coding was performed: line-by-line, assigning labels to meaningful units of text (words, phrases, or exchanges that conveyed a distinct idea). After familiarization with the transcripts, both researchers independently conducted open coding. Codes were compared and refined into a shared codebook that guided subsequent axial coding. Examples of initial codes included “loss of routine”, “community invisibility”, “fragmented support”, and “anticipatory grief”. Substantial overlap was found in the coding, and differences were resolved by revisiting the transcript and refining code definitions collaboratively. The analysis follows an interpretive, reflexive thematic analysis approach [32], in which consensus-building and reflexive dialogue are emphasized rather than statistical inter-rater reliability. The focus was to develop a rich, conceptually grounded framework through investigator triangulation and consensus. In line with reflexive thematic analysis, we did not calculate a statistical inter-rater reliability index (e.g., Cohen’s kappa). Instead, rigour was ensured through investigator triangulation: both researchers independently coded the first transcript, compared and discussed their coding, and resolved discrepancies through consensus, refining the codebook iteratively. This process was repeated for subsequent transcripts to enhance credibility and dependability.

Themes were developed when multiple related codes clustered meaningfully and could be defined by a coherent central organizing concept. Codes were compared and refined into a shared codebook that guided subsequent axial coding. Criteria for theme development included: (a) recurrence across focus groups; (b) conceptual relevance to the research questions; (c) richness of supporting excerpts; and (d) coherence and distinctiveness from other emerging categories. This was done independently for the first focus group transcript and repeated for the second and third focus group transcripts.

Axial coding and clustering of codes were then conducted to identify broader themes and sub-themes. Cluster related codes were used according to explicit criteria: recurrence across groups, relevance to post-caregiving experiences, and conceptual connection to community-based support. For example, codes such as “feeling discarded”, “compassion fatigue”, and “lack of public recognition” were grouped into the Compassion Axis. “Need for follow-up”, “absence of structured pathways”, and “poor communication between services” were clustered into the Coordinated Action Axis, while “lack of training”, “navigating rights”, and “need for narrative competence” formed the Care Literacy Axis. Unique perspectives that appeared only in one group but were particularly salient were also noted. To enhance the narrative depth of the analysis, a Narrative Medicine lens was integrated by attending to metaphors (e.g., “a net with holes”), emotional tone (such as frustration, hope, etc.) and implicit values (such as compassion, dignity, and recognition), which helped refine themes through meaning-making rather than frequency alone. Interaction analysis [32] was incorporated to capture how participants influenced each other, particularly where consensus or divergence shaped theme credibility. Changes in participants’ stances during the conversation were also noted, as these can indicate shifts in perspective due to the discussion.

After synthesizing the findings, three major thematic axes emerged that encapsulated the ideas needed for a post-caregiver support network: Compassion, Coordinated Action, and Care Literacy. These themes appeared interlinked under the umbrella concept of Community-For-Care. A thematic map (see Figure 1 in Results) was developed to illustrate how the overarching theme and sub-themes relate. The raw data were revisited to ensure that each theme was well-supported by participants’ quotes and that the analysis captured the essence of participants’ contributions without imposing researchers’ own preconceptions. Exemplar quotes from participants (anonymized, labeled P1, P2, etc.) were included to highlight key points in their own words; these quotes were translated into English for reporting (the focus groups were conducted in Portuguese).

Trustworthiness of the analysis was enhanced through several measures: (1) investigator triangulation—both authors analyzed and agreed upon the coding and themes; (2) inclusion of rich verbatim extracts in the report to allow readers to see evidence of the themes; (3) informal member checks—a summary of the findings was returned to two participant representatives (one health professional and one post-caregiver) to ask if the themes resonated with their experience. Both indicated that the findings made sense and accurately reflected the discussions they participated in, providing some confidence that the interpretations aligned with participants’ intent.

### 2.5. Ethical Considerations

This study was conducted in accordance with ethical standards for research involving human participants. The project received approval from the Ethics Committee of Fernando Pessoa University, approval number: [393/23/], prior to commencement. All participants provided informed consent. Given the online nature of the focus groups, consent was obtained electronically: participants received the information sheet and consent form via email, which they signed and returned. At the start of each session, the moderator verbally confirmed consent to continue and to record the discussion.

Participants were assured of confidentiality and anonymity. It was emphasized that any quotes or examples used in publications would be stripped of identifying details. Each participant was assigned a code (P1, P2, etc.) for note-taking and reporting. The recordings were stored securely on a password-protected device accessible only to the research team and were deleted after transcription was completed and verified. Transcripts were labeled only by focus group number and participant code, without names.

The research team remained attentive to the emotional sensitivity of the topic, acknowledging that discussions about caregiving and the post-caregiving period could evoke grief or distress among participants, including professionals. In accordance with the ethical protocol, the moderator was instructed to pause the discussion when necessary, acknowledge participants’ emotional responses, and resume only if they felt comfortable to proceed. Following each focus group, a debrief email was sent to all participants, including a note of appreciation, information on caregiver support and bereavement counselling services, and contact details for further inquiries. One participant (a post-caregiver) reached out afterward to thank the team, mentioning that she found the group discussion therapeutic, as it was the first time she felt her post-care experience was heard. No participants reported harm or requested any form of formal psychological follow-up due to the study, but the inclusion of support information was an important precaution.

The study adhered to the principles of the Declaration of Helsinki. Participants were explicitly informed that involvement was voluntary and that they could withdraw at any time without any consequences. It was clarified that declining or discontinuing participation would not affect any services or relationships (for example, with healthcare providers or associations). All data were handled in compliance with data protection regulations; only aggregated findings are presented, and individual identities remain confidential.

Finally, to ensure rigor and transparency, an audit trail of the research process was maintained, including documentation of how the focus group guide was developed, decisions made during coding, and minutes of analysis meetings. This provides clarity on how interpretations were reached and allows others to assess the credibility of the findings.

## 3. Results

Participant Characteristics and Group Dynamics: A total of 15 participants (12 women, 3 men) took part in three focus groups. Their ages ranged from 35 to 70 years. Participants represented a spectrum of roles related to caregiving. All focus groups were lively, with high engagement. Notably, the interaction analysis did not reveal any single individual dominating the discussions, nor obvious hierarchies based on professional status; for instance, in the first focus group of health professionals, both doctors and nurses/caregiver representatives took turns leading parts of the conversation. In the mixed groups, mutual respect was observed—professionals listened attentively to caregivers’ personal accounts, and caregivers in turn showed interest in professionals’ viewpoints. This dynamic suggests that the topic—support for post-caregivers—was approached by all as a shared concern rather than a contentious issue. In the presentation of findings below, participant quotations are shown as separate, indented excerpts to enhance readability, and it is indicated when a theme was shared across all groups or was more specific to a particular stakeholder subgroup.

Across the groups, participants converged on a central idea: the community itself needs to be the main pillar of support for informal post-caregivers. This was a cross-cutting theme present in all three focus groups. One participant metaphorically described the current support system as “*a net with holes*” that post-caregivers fall through, and said the solution is “activating the living forces of the community to weave those holes shut”. The term “Community-For-Care” emerged organically in one discussion and became a unifying theme in the analysis. Community-For-Care is defined as a community-driven support network devoted to caring for those who have finished caregiving. Participants envisioned this as a holistic approach rallying local resources—people, knowledge, services, compassion—around the post-caregiver.

Through qualitative analysis, three major thematic axes were identified that constitute the foundation of the Community-For-Care network: Compassion Axis, Coordinated Action Axis, and Care Literacy Axis. These were consistently mentioned and strongly interlinked in participants’ narratives. Figure 1 illustrates the thematic map of the proposed Informal Post-Caregiver Support Network, showing Community-For-Care at the center (the overarching framework) with the three axes surrounding it and feeding into one another.

Figure 1 illustrates the Community-For-Care framework as an integrated, dynamic system composed of three interdependent axes, Compassion, Coordinated Action, and Care Literacy, organized around the central concept of a community-driven support structure for informal post-caregivers. The bidirectional arrows indicate that these axes do not operate in isolation; instead, each axis functionally reinforces and enables the others, creating a cyclical and mutually amplifying relationship. Technically, the Compassion Axis provides the affective foundation that mobilizes community engagement and social recognition of post-caregivers. This heightened community sensitivity increases the likelihood that coordinated mechanisms are activated—facilitating smoother referrals, follow-up pathways, and cross-sector collaboration (Coordinated Action Axis). In turn, a coordinated network creates the conditions for sustained educational initiatives and information flow (Care Literacy Axis), ensuring that caregivers, post-caregivers, and professionals possess the knowledge necessary to engage meaningfully with the system. Care Literacy then feeds back into the other axes by equipping community members and professionals with the competencies required to act compassionately and to participate effectively in coordinated structures. When literacy increases, compassionate behaviours become more intentional and widespread, and coordination becomes more efficient and evidence-informed. Thus, the axes interact as a self-reinforcing cycle: compassion motivates action, coordination structures the action, and literacy empowers all actors to participate, ultimately producing a robust and adaptive community support network. The figure therefore represents an ecological model in which each axis is both a driver and a product of the others, illustrating how Community-For-Care functions as a systemic and dynamic framework rather than a set of isolated components.

### 3.1. Community-for-Care: An Overview

The community (“living forces of the community”) is at the core, symbolized by the central circle. Surrounding it are three interrelated axes essential for supporting informal post-caregivers: A Compassion Axis (fostering a culture of care and empathy), a Coordinated Action Axis (integrating efforts of healthcare, social services, and community actors), and a Care Literacy Axis (educating and empowering caregivers, post-caregivers, and professionals). Participants across all groups stressed that community involvement is both the context and the engine of any support system for post-caregivers. The community was described as the “*village that cares*”, invoking the proverb “*it takes a village*”—in this case, to care for the caregiver who has cared for someone else. The idea is that when the formal caregiving role ends, the community collectively should step forward to care for the post-caregiver. As one participant (P2 a community association member) put it, “*We talk a lot about communities of care for patients, but we also need a community for the informal caregivers—a Community-For-Care that doesn’t abandon you after the funeral*”.

Community-For-Care implies re-centering support at the community level, rather than expecting individual post-caregivers to navigate siloed services or cope entirely on their own. It encompasses emotional support (neighbors checking in, peer support groups), practical help (community volunteers providing respite or assistance with errands), and inclusive social structures (ensuring post-caregivers are invited and welcomed in community events and dialogues, not left isolated). Participants noted that communities already have latent “*living forces*”—compassionate individuals, local leaders, faith groups, social clubs, etc.—that could be harnessed more effectively to create a safety net. This perception of “latent community resources” was particularly emphasized by members of caregiver associations and civil society organizations.

The three axes below detail the key dimensions participants identified within this community-centric model.

### 3.2. Compassion Axis

The Compassion Axis refers to cultivating a culture and value of compassion throughout the community, so that caring and empathy become normative “*reflexes*” in society’s response to post-caregivers. Participants diagnosed a lack of recognition and empathy fatigue in society regarding informal caregivers, which naturally extends to post-caregivers. Many felt that caregiving is undervalued—an invisible labor taken for granted—and that once a caregiver’s duties end, society assumes they are simply “*free*” to resume normal life. This axis, therefore, is about raising awareness and sensitivity so that communities appreciate the caregiver role and its aftermath. The need for greater social recognition of post-caregivers was a theme raised in all three groups, and it was especially vivid among post-caregivers themselves.

Participants shared heartfelt accounts illustrating the current compassion gap. One post-caregiver (P8) described the feeling of being discarded after her care recipient died: “*We [post-caregivers] are very little valued for the work we did and the needs we still have. The role of the caregiver is highly undervalued at all levels. It starts with being devalued by health professionals, who use us as a resource to fill gaps in services … Society in general only understands the caregiver’s role when they go through it themselves. When you stop being a caregiver, no one is interested in whether you’re physically or psychologically unwell. It’s like, the next day you’re expected to go back to work as if your problem is over*” (P8, post-caregiver).

This quote highlights a perceived dismissal of the post-caregiver’s experience: their years of labor and sacrifice aren’t formally acknowledged, and their subsequent struggles (health issues, grief) are ignored or trivialized. P8’s mention of health professionals viewing family caregivers as a “*mere resource*” also points to systemic issues—caregivers often fill in for inadequate home care services, yet their partnership is not reciprocated with support for them later on.

Another participant (P3, a palliative care psychologist) emphasized the emotional toll on caregivers and how that continues post-care: “*Family caregivers are very lonely and need to talk. I see them [during caregiving] going through several bereavements, because they are losing the person in pieces—losing various abilities and identities sometimes—long before the final bereavement of death*” (P3, psychologist).

This underscores that by the time caregiving ends, individuals may have already endured prolonged anticipatory grief and serial losses, leaving them emotionally depleted. Yet the community often does not acknowledge this cumulative grief. The Compassion Axis would involve validating these experiences and ensuring post-caregivers feel seen and heard.

One concrete strategy suggested to bolster compassion was the creation of a platform for sharing caregiving experiences. Participants imagined an online (and/or community center-hosted) platform where current and post-caregivers could narrate their stories—the challenges, the joys, the lessons learned. By doing so, these personal stories could resonate with others and “put a face” to the caregiving journey for the broader public. As P4 (caregiver association representative) described: “*We need something like a digital wall of stories. It starts small, like a snowball—you post one story here, one there, multiply the actions, load the platform with up-to-date testimonials. It gives meaning to those who share, and it educates those who read*” (P4, association representative).

The rationale for the platform is twofold: for the post-caregivers or caregivers who share, it is therapeutic and validating (making meaning of their experience through writing, as narrative medicine advocates); for the readers (community members, policymakers, health professionals), it builds empathy and could drive changes in attitudes or policies. Participants noted that positive caregiving experiences should be shared, not just negative ones, so that society can also see caregiving as valued work and not solely a burden. In essence, if caregiving is often described as rewarding yet challenging, those rewards—the love, personal growth, human connection—should also be highlighted to build a collective appreciation for care in the community ethos.

In discussing compassion, the idea of the “*village of care*” emerged. This envisions a community where caring is integrated into daily life, not something that only happens in crisis. In a compassionate community, neighbors might routinely check on each other’s well-being, people are attentive to who in their vicinity might be a caregiver or post-caregiver, and local culture prizes kindness. Some participants cited examples like community-organized visiting programs for widows or support circles in churches that rally around bereaved families. The consensus was that these organic acts of compassion need scaling and formal encouragement. A participant (P7, caregiver support institution member) noted:

*“The value of caring, as a shared value, creates a reflexive, cascading movement. If we as a community truly value caring—like celebrate caregivers at events, talk about it in schools, whatever—it will open space for everyone to reflect on how we look after each other. Particularly how we treat our elders and those exhausted from caring for them”*.(P7, support institution member)

This passionate statement calls out ageism and dependency stigma—implying that a compassionate community does not discard people who are old, dependent, or who have “*fulfilled their duty*” and are now themselves in need (like many post-caregivers). Thus, compassion here is linked to social inclusion and dignity.

In summary, the Compassion Axis involves raising community consciousness to truly see informal caregivers and post-caregivers, valorizing their roles, and fostering an environment where asking for and offering help is normalized. Strategies include storytelling platforms, awareness campaigns (e.g., local pamphlets or social media highlighting “Caregiver of the Month” stories), and community ceremonies or acknowledgments (some participants suggested an “Informal Caregivers Day” celebration). The expected outcome of strengthening this axis is a more empathetic society that organically provides emotional support and reduces the isolation of post-caregivers.

### 3.3. Coordinated Action Axis

The Coordinated Action Axis focuses on the need for different services, sectors, and actors to work in a concerted, continuous way to support caregivers through the entire caregiving trajectory—including after caregiving ends. Participants universally criticized the current support system as fragmented and reactive. They pointed out that while some resources exist (like caregiver allowances, respite care options, bereavement counseling), these are often disjointed, hard to navigate, or temporary. There was a call for a more integrated approach that would seamlessly accompany an individual from being a caregiver to becoming a post-caregiver. This theme was voiced in all three focus groups, with particularly detailed examples from health professionals and caregiver association members.

One participant (P1, a palliative care physician) described the status quo bluntly: “*The system is highly fragmented, it’s not integrated, and the responses are too scarce and one-off. They’re often based on charity or short-term relief instead of a sustainable plan*” (P1, physician).

This critique aligns with literature that advocates for continuous caregiver support programs rather than episodic interventions. P1 emphasizes that many current efforts are not sustained or scaled (e.g., a time-limited project in one locale that doesn’t continue or expand). Participants suggested that coordination needs to occur on multiple levels:

Healthcare integration: Ensuring that when a patient under care dies or is institutionalized, the healthcare system does not simply close the case, but actively transitions the caregiver into follow-up support. One strong proposal was to establish a “Caregiver Consultation” or clinic within the healthcare system. For instance, P9 (a nurse) suggested a dedicated consultation for caregivers and post-caregivers: “*Why do we have a well-baby check-up after birth, but nothing analogous after a death? There should be a consultation for the caregiver—integrated into hospital discharge planning or in primary care—that specifically looks after the caregiver’s health and plans the next steps for them*” (P9, nurse).

This idea implies a systematic change: including caregivers in care plans of patients and formally handing them over to a post-care program upon the patient’s death or placement in a facility. Such a program might involve health screening for the post-caregiver (many neglect their own health), mental health support, and linkage to community resources.

Cross-sector collaboration: Participants identified that multiple sectors have roles—healthcare, social services, local government, NGOs, religious organizations, employers (for working caregivers), etc. A coordinated network could involve appointing a case coordinator or a “*community care navigator*” who helps connect a post-caregiver to various supports (grief counselors, financial advice, social benefits, volunteer help). P5 (a general practitioner) mentioned the importance of social and mental health support alongside medical: “*Healthcare must be designed to provide support over time according to the caregivers’ evolving needs—not just financial support (though that’s important), but real reintegration support. Right now in our legislation there’s basically nothing after the caregiving ends. We need good mental health services, social work follow-up, maybe job re-training for those who can still work. It has to be a package that is coordinated—primary care, mental health, social security, all talking to each other*” (P5, a general practitioner).

This suggests setting up protocols where, for example, when a death is registered or a palliative care service discharges a patient due to death, an alert could go to a community care coordinator who then proactively contacts the family caregiver to offer support services.

Persistent timeline: A crucial element is longitudinality—participants felt support shouldn’t cut off shortly after bereavement, but continue as needed for months or years. Some suggested a tapering model: intensive support in the first weeks (e.g., daily check-ins by a community health worker or volunteer, help with funeral and paperwork), then regular check-ins for a year (monthly calls or support group meetings), and easy re-entry if help is sought later. P10 (a post-caregiver who is now an association volunteer) described the abrupt cutoff:

*“It’s extremely violent how from one day to the next, the caregiver’s entire world disappears. The healthcare team is gone, the routine is gone. One day you’re giving 110%, the next day you’re not needed. That transition from full throttle to dead stop has to be prepared in advance”*.(P10, post-caregiver)

This quote illustrates the emotional whiplash of transition and underscores the need for a planned handover. In a coordinated system, as P10 notes, preparing the caregiver for the end should be part of end-of-life care practice: healthcare professionals can, while the care recipient is nearing the end, start discussing with the caregiver what will happen afterward (both practically and emotionally), perhaps referring them to resources preemptively.

In terms of who should coordinate, participants had various ideas. Some felt primary care (family doctors and community nurses) are best positioned because they have ongoing relationships with families and a mandate for continuous care. Others thought specialized palliative care teams or hospice programs might extend their care to the family for a period post-death (some palliative care services internationally provide bereavement follow-up for families for up to a year; participants wished this were more widespread in Portugal). There was also mention of municipal roles—local councils could create “*Caregiver Support Offices*” to centralize information and referrals, ensuring no one falls through the cracks.

Another component of coordination is building partnerships with community organizations and volunteers. This ties back to compassionate communities: for instance, a local volunteer network (through a church, Red Cross, or a Compassionate Community initiative) could be on call to provide home visits to a new widower to reduce loneliness, or drive a post-caregiver to social events to help reintroduce them to community life. Participants suggested that formal services should coordinate with such informal support: e.g., a doctor or social worker, with consent, could refer a post-caregiver to a community volunteer group for companionship.

One novel suggestion was establishing a post-caregiver peer support network where post-caregivers support newer post-caregivers (the idea that “*only someone who’s been through it really gets it*”). This could be coordinated by an NGO or community center. Given that participants like P10, who had become a volunteer, exist, harnessing their empathy and knowledge in a structured way could help newly bereaved caregivers. This peer network could operate like a mentoring or buddy system.

To illustrate the Coordinated Action Axis, consider a hypothetical but typical scenario raised in the focus group: A 60-year-old woman cared for her husband with cancer for 5 years. He passes away at home under palliative care. In a coordinated model, the palliative care nurse and social worker would meet with her shortly after, provide condolences and a “*post-care info pack*” (including contacts for grief counseling, legal/financial advice, support groups). Her family doctor is alerted to do a check-up within a month. A volunteer from a local compassionate community program might start visiting weekly or invite her to a “*tea time*” for widows. A few months later, a follow-up call is made by a caregiver support office to see how she’s coping and whether she needs help returning to work or engaging in community activities. All these moving parts require planning and communication across agencies—exactly what the Coordinated Action Axis encapsulates.

Participants acknowledged that implementing such coordination is challenging and requires political will and resources. They argued, however, that the payoff would be significant: healthier, more resilient post-caregivers who are less likely to suffer severe health decline or require institutional care themselves. In the long run, this could even be cost-saving for the healthcare system. It was evident that participants viewed the current lack of coordination as a major weakness that the Community-For-Care model must fix. Such proposals tended to be more elaborated by professionals and organizational representatives, whereas post-caregivers focused more on describing the lived consequences of fragmentation (e.g., loneliness, sudden loss of routine).

### 3.4. Care Literacy Axis

The Care Literacy Axis involves education, training, and knowledge-sharing for all parties involved in the caregiving and post-care process—including caregivers and post-caregivers themselves, health professionals, and the community at large. By improving “literacy” in care (both the practical skills and the understanding of the caregiving journey), individuals can be empowered and the support network made more effective.

Several needs were articulated under this theme:

Training and information for informal caregivers/post-caregivers: Participants argued that one reason caregivers struggle is lack of preparation and knowledge about both caregiving tasks and self-care. Thus, by extension, many post-caregivers might have coped better if they had better training during their caregiving period. A specific idea was a centralized training platform or repository of caregiver resources. P7 (a community health worker) said: “*We need a platform that promotes literacy about the rights and resources for caregivers, and even basic know-how. If they were more empowered and taken care of during caregiving, they’d emerge as post-caregivers in better mental and physical health, not completely burned out*” (P7).

This platform could aggregate tools and resources from different caregiver associations, covering topics like caregiving techniques (safe mobility, medication management), legal/financial guidance, respite options, self-care strategies, and what to expect when caregiving ends (mourning, returning to work, etc.). It should be user-friendly for a general audience. One benefit of an online platform is broad accessibility; however, participants also noted the importance of in-person literacy efforts: workshops, community meetings, etc., for those less tech-savvy or to build personal connections.

Psychological literacy and bereavement support: Many participants emphasized educating caregivers and families about the emotional aspects of caregiving, including anticipatory grief and bereavement. If caregivers had more knowledge about the grieving process, they might be better prepared for the waves of emotions post-care. Some suggested offering bereavement literacy sessions—possibly integrated into hospice or hospital bereavement programs—where caregivers approaching end-of-life situations could learn about common feelings after loss, coping mechanisms, and the fact that seeking help is okay. P11 (a psychologist) stressed that professionals themselves need this training: “*The truth is health professionals don’t receive training in grief and bereavement during their studies. We need to incorporate this into training at all levels—primary care, hospital teams, even in medical/nursing school. How to care for the bereaved caregiver, how to identify complicated grief*” (P11, psychologist).

This points to a gap in professional training. By increasing healthcare providers’ literacy in supporting caregivers and post-caregivers, the formal system can respond more appropriately (instead of leaving it all to chance or the community).

Empowering post-caregivers through knowledge: Some participants advocated for specific skill-building for post-caregivers to help them reintegrate into society. For instance, if a caregiver left the workforce and is at risk of long-term unemployment, providing them with adult education or job training is a form of literacy (economic/skills literacy) that could be transformative. While not traditionally framed as “care literacy”, it was mentioned that supporting post-caregivers might include helping them gain new competencies or certifications to re-enter jobs, or offering financial planning advice if their economic situation changed after caregiving. In other words, part of caring for post-caregivers is equipping them for their next chapter.

Streamlining information about rights and benefits: The relatively new caregiver law in Portugal and existing benefits (like respite care or caregiver allowance) are often not widely known or are confusing to navigate. Participants recommended that the support network include clear guidance on “*what am I entitled to, and how do I get it?*” as a key literacy component. This could be through informational brochures, a website, or trained personnel (nurses, social workers) who proactively educate caregivers.

These literacy-related needs were described across all groups, but professionals tended to focus on curricular and institutional changes, whereas caregivers and post-caregivers emphasized practical information and peer-to-peer exchange.

Concrete proposals under this axis included:

Developing curricula for caregiver education (potentially delivered by healthcare providers or NGOs) that could be offered when someone registers as an informal caregiver under the new law. This would ensure a baseline knowledge, as one participant said, “like prenatal classes, but for caregiving”.

Organizing community literacy events (health fairs focused on caregivers, public lectures on aging and caregiving, etc.) to raise general awareness. This ties back to the Compassion Axis because as the public becomes more literate about what caregiving entails, compassion increases.

Encouraging peer knowledge exchange: e.g., experienced caregivers mentoring new caregivers, or post-caregivers sharing tips on how they coped with certain transitions. Participants noted that often practical tips (like how to get some sleep while caregiving, or how to deal with paperwork after someone dies) are not found in manuals but can be obtained from those who’ve been through it.

Notably, some participants (especially from caregiver associations) felt that they were doing a job that should be done by formal entities. For example, one association representative said they spend a lot of time informing caregivers about basic rights that hospitals or social security offices should have told them. This indicated a systemic gap in literacy dissemination by official channels, hence NGOs and communities stepping in. The Care Literacy axis in the Community-For-Care model would ensure that such information is centralized and consistently provided, ideally with institutional support rather than solely volunteer-run.

Finally, raising literacy among health professionals was strongly emphasized. Beyond bereavement training, participants said professionals need to recognize caregivers (and by extension post-caregivers) as part of the care unit. Simple acts like a doctor asking a caregiver “*How are you holding up?*” during a patient visit can make a difference, as can giving advice on self-care or referrals to caregiver support. If professionals are literate in caregiver issues, they can educate patients’ families early, refer them to resources, and follow up appropriately.

In summary, the Care Literacy Axis complements the Compassion and Coordination axes by ensuring everyone has the knowledge and skills to effectively participate in the support network. It empowers caregivers and post-caregivers through information, prepares professionals to engage with them empathetically and effectively, and informs the community so that caregiving is a visible, understood part of life.

Taken together, the three axes form a synergistic framework: compassion (reported across all groups) motivates action, coordination (particularly elaborated by professionals and organizational actors) structures that action over time, and literacy (highlighted by both lay and professional participants) equips all stakeholders to participate meaningfully. This interdependence underpins the Community-For-Care concept as an integrated, community-anchored model for supporting post-caregivers.

## 4. Discussion

This study set out to explore how an integrated community-based network could support informal post-caregivers, and the findings coalesced around the concept of Community-For-Care, underpinned by compassion, coordinated action, and care literacy. In discussing these results, it is useful to situate them in the context of existing literature and frameworks, and to consider their practical implications and challenges. Rather than adding a new label alone, the present study specifies how these three axes operate together as an applied, post-caregiver-focused extension of existing community approach.

Firstly, the emphasis on the community as the central pillar of support aligns with the growing movement of compassionate communities in health and social care. Compassionate community approaches advocate that care for people experiencing serious illness, caregiving, dying, or loss is best supported when the community at large is engaged and responsive. This is consistent with findings by Larkin [18] and others who have noted that post-caregivers benefit from community recognition and inclusion in social activities as they rebuild their lives [20,21,22]. The Community-For-Care model advances this literature by making explicit that the primary “unit of intervention” is the post-caregiver in transition, and by articulating community roles that extend beyond generic bereavement support to address identity reconstruction and social re-engagement after caregiving.

The Compassion Axis identified in this study can be viewed as a localized application of the compassionate community philosophy. It is about creating a culture of care at the micro (neighborhood) and meso (municipal) levels. This is not necessarily a quick or easy task—it involves cultural change and sustained public engagement. However, programs around the world provide models: for instance, Compassionate City charters (such as those in the UK, Canada, and in some cities in Portugal) explicitly aim to raise public awareness about caregiving and loss, encouraging citizens to take action (like committing to check on people who are bereaved, or to support caregiver respite) [29,30,33,34]. By referencing the need for compassionate communities and explicitly naming compassion as an axis, this study adds to the empirical support for these initiatives. At the same time, Community-For-Care specifies concrete compassion-building mechanisms (e.g., narrative platforms, public recognition, community rituals) that translate a broad ethos into practice for a clearly defined population—informal post-caregivers. These data support the idea that compassion alone is not enough—it must be channeled into structured support (hence the Coordinated Action axis) and informed by knowledge (Care Literacy axis). This helps address a critique sometimes levied at compassionate community approaches: that they can be abstract or “feel-good” but need concrete implementation pathways. The participants’ suggestions (story-sharing platforms, awareness campaigns, community recognition events) provide tangible avenues to operationalize compassion. Here, the three axes operate as such a pathway, linking values (compassion), structures (coordination), and capabilities (literacy) in a single framework.

The Coordinated Action Axis highlights a well-known challenge in caregiving support: fragmentation of services. The transition from caregiving to post-caregiving can slip through cracks between healthcare, social care, and community services. The findings strongly advocate for a continuum-of-care approach, effectively extending the care pathway to include the caregiver’s journey after the formal caregiving ends. This aligns with the concept of “integrated caregiver support” proposed by some researchers and policy advocates. Caregiver Identity Theory implies that interventions should be matched to where caregivers are in their “career” [35,36,37]. In early caregiving, support might focus on role acquisition; in late caregiving, on anticipatory grief and separation; and in post-caregiving, on identity reconstruction and reconnecting with prior roles. A coordinated network would adapt to these phases seamlessly [38,39]. The contribution of this study is to translate these theoretical propositions into a set of practice-oriented elements—such as post-caregiver consultations, systematic follow-up after a patient’s death, and community care navigators—that could be embedded in existing health and social care structures. These observations support such an approach—participants implicitly described the need for phase-specific support (training and respite during caregiving, bereavement counseling and community re-engagement after caregiving).

The results lend weight to arguments that such legislation and policies should explicitly incorporate post-caregiver support, such as guaranteed follow-up calls or health checks, transitional financial support, or automatic enrollment in bereavement support groups. The idea of a post-caregiver consultation or clinic is particularly innovative and could be piloted in healthcare systems. A parallel can be drawn to how obstetric care includes a postnatal check-up for mothers—by analogy, palliative care or primary care could include a “post-caregiving check-up” for family caregivers a few weeks after a death. This could be an occasion to assess the post-caregiver’s health (mental and physical), offer guidance, and connect to community resources. Given that caregivers often neglect their own health, this would be an opportunity to catch issues early. In policy terms, the Community-For-Care model therefore suggests a concrete agenda: integrating post-caregiver follow-up into discharge and bereavement protocols, and mandating formal links between clinical services and community-based resources.

Another aspect of coordination is communication between acute care, hospice, and primary care. Participants recommended better communication at the time a care recipient’s life ends. A practical implication is developing protocols where hospitals or hospice programs notify primary care providers of a patient’s death and identify the primary caregiver. That way, primary care can follow up with the caregiver. Such coordination requires systemic change but could likely be implemented via electronic health record alerts or inter-agency agreements. At the same time, institutional constraints must be acknowledged: workload pressures in primary care, variability in palliative care coverage, and uneven digital infrastructure may limit the feasibility of universal follow-up, particularly in resource-constrained settings. Implementing Community-For-Care will thus require prioritization, phased roll-out, and alignment with existing regional or national strategies for informal care.

The Care Literacy Axis ties into a broader discourse on empowering caregivers. Studies have shown that caregiver outcomes improve when they feel confident and knowledgeable in their role. For example, training programs for dementia caregivers have been linked to reduced burden and improved coping. However, most training ends at the caregiving phase—the findings suggest extending education into the post-care phase as well. For instance, offering short courses or counseling sessions on “Life after caregiving: what to expect and how to navigate it” could be very beneficial. This might include content on normalizing feelings of grief or relief, strategies to rebuild one’s social network, and how to take care of one’s health post-care. The peer-led dimension is significant: post-caregivers can be excellent educators for current caregivers, providing real-life tips and empathetic support. Peer support interventions (especially in the context of dementia care) have had promising results for reducing isolation and improving mental health among caregivers. Extending these into bereavement support could harness that same peer empathy for post-caregivers. Within the Community-For-Care framework, literacy is therefore conceptualized not only as clinical or procedural knowledge, but also as narrative, emotional, and socio-economic literacy that helps post-caregivers navigate work, income, and identity transitions.

Participants specifically highlighted a gap in professional training regarding bereavement and caregiver support. This echoes literature in palliative care where many clinicians report feeling ill-equipped to handle conversations about grief or to support families beyond the patient’s death. Improving professionals’ narrative competence—their ability to listen to and acknowledge caregivers’ stories—could make healthcare encounters more supportive for caregivers and post-caregivers. For example, if a year after a patient’s death a post-caregiver comes to a clinic with stress symptoms, a narrative-trained clinician might pick up on their caregiver history and explore how that experience is affecting them, rather than just treating it as an isolated medical complaint. In this sense, the study shows how Narrative Medicine can be operationalized at system level: not only in individual encounters, but as a guiding ethos for training, supervision, and reflective practices that sustain Community-For-Care over time. The findings extend previous research on post-caregivers in a practical direction. Earlier studies identified what post-caregivers go through (grief, identity loss, health issues) [20,21,22]. This study contributes by suggesting how to respond—via a community-centric network with specific components. This moves the conversation from solely descriptive to solution-oriented. In doing so, concepts from public health (compassionate communities), psychology (transitional stages, identity reconstruction), and education (literacy empowerment) are integrated. Taken together, the three axes clarify where action should occur (in and with communities), who should be involved (post-caregivers, professionals, associations, local authorities), and what the priorities should be (recognition, continuity of support, and shared knowledge).

One might view Community-For-Care as a person-centered, community-driven extension of the palliative care philosophy. Palliative care has long championed caring for families and bereavement follow-up, but often these remain within the remit of specialist teams. This model suggests radiating that ethos outwards: empowering the community to take up some of that mantle, with professionals as supporters or facilitators. Future work will need to test the model under real-world constraints—limited funding, variable community engagement, and competing policy priorities—and to identify which components are essential, which can be adapted, and what level of investment is required to make Community-For-Care sustainable at scale.

## 5. Limitations

The sample size was small (15 participants) and selected through non-random, purposive methods. Participants were recruited through caregiver associations, healthcare teams, and community organizations. Because several participants were affiliated with caregiver associations or community support groups, and may therefore be more motivated, engaged, or sensitized to caregiver issues than typical post-caregivers, the sample may not represent the full diversity of experiences. This may have introduced a bias toward an “activist” perspective, particularly in emphasizing community responsibilities and systemic gaps. Additionally, the sample was predominantly female (12 out of 15 participants) and largely drawn from rural or semi-rural settings, which may influence the types of community-based support envisioned and the perceived accessibility of resources. It is possible that other caregivers or health professionals who did not participate might have different perspectives or experiences. Therefore, the findings are not broadly generalizable in a statistical sense.

The focus group method itself also has limitations. Group discussions can lead to polarization or conformity. If one participant stated a strong opinion (e.g., “the government should do X”), others might have felt pressure to agree or not voice counter-opinions, especially if there were perceived hierarchies (like a respected doctor speaking up). Efforts were made to minimize power dynamics by grouping similar categories together and by explicitly inviting quieter members to share. In the analysis, a high degree of consensus on major points was noted—while this strengthens those points, it also suggests caution if that consensus was partly a product of the group context. It’s also possible that some participants were guarded in their comments due to the presence of others. For example, a post-caregiver might not want to seem critical of healthcare in front of professionals, or vice versa. Although participants were candid about system flaws, a truly uninhibited critique might be more likely in a private interview. Therefore, the data may underrepresent conflictual or highly negative experiences. Interaction effects are inherent to focus groups; the researchers did their best to capture them (through interaction analysis) but some nuances might have been missed.

The nature of the topic—revisiting personal caregiving and loss experiences—carries emotional weight. There is a possibility of recall bias or emotional bias in how participants recounted their experiences or ideas. Moreover, participants were at different stages of the post-caregiving trajectory, and time since caregiving varied widely. This temporal heterogeneity may have shaped the narratives: individuals still in early bereavement may focus on emotional needs, while those further along may emphasize identity reconstruction or practical reintegration. These different time perspectives all converged in the focus group conversation. The analysis did not stratify based on how long ago participants had been caregivers, which could influence their viewpoints. Future studies may benefit from distinguishing between early and later post-care phases to better understand evolving support needs.

In presenting the Community-For-Care model, the authors are aware that it has been conceptualized as a relatively idealized structure based on themes. The data, while rich, are still preliminary in that no extensive evidence was gathered on how each suggestion would work in practice or how effective it would be. For example, the idea of a storytelling platform is appealing, but there are no data on how many post-caregivers would actually use it, or if reading others’ stories demonstrably increases community empathy. The recommendations should be seen as hypotheses or proposals that require further pilot testing and evaluation. Similarly, the “axes” are an interpretative framework—in reality, there could be other ways to categorize the needs (someone else might frame them as emotional, informational, and instrumental support needs, which overlaps with this model but is a different taxonomy).

Despite these limitations, the study offers valuable insights. It provides a proof of concept that engaging multiple stakeholders can yield a comprehensive vision for supporting post-caregivers. The consistency of certain themes across different groups (professionals and caregivers alike) suggests that the issues identified are salient and real. Additionally, triangulation with the literature (where the findings resonated with known issues like identity loss, need for grief support, etc.) adds credibility. Nonetheless, the limitations indicate that more research is needed, ideally of various designs: qualitative studies in other settings to compare themes, quantitative surveys to gauge how common certain needs or opinions are, and intervention studies to test some of the proposed solutions.

## 6. Conclusions

This study highlights the need for a structured, community-anchored response to the challenges experienced by informal post-caregivers. Unlike conventional compassionate community models, Community-For-Care targets the specific transition following the end of caregiving. The framework proposes operational pathways that can be translated into practice.

The Community-For-Care model offers a novel, post-care-specific extension of compassionate community principles, articulating three operational axis: compassion, coordinated action, and care literacy. Together, they define a pathway for structured community support after caregiving ends. Rather than reiterating general calls for social support, this model clarifies how community actors, services, and knowledge resources can interconnect to sustain post-caregivers during identity transition, grief, and reintegration into daily life. Building on these findings, concrete steps for implementation can be envisioned. A feasible next phase would be the development of a pilot Community-For-Care program in one municipality. Such a pilot could include:(1)A Community Compassion Hub, responsible for awareness campaigns, storytelling initiatives such as a platform for post-caregivers, and volunteer mobilization;(2)A Post-Caregiver Coordination Pathway, connecting healthcare, social services, and local organizations through standardized referral mechanisms and scheduled follow-up contacts after caregiving ends;(3)A Care Literacy Platform, integrating training for caregivers and post-caregivers, educational modules for professionals, and clear guidance on rights and resources.

Process and outcome indicators, including post-caregiver well-being, social reintegration, service utilization, and community engagement, could be monitored and compared with a similar community without the program.

Future research should test and refine these components across different sociocultural settings, examine their long-term impact on post-caregiver health trajectories, and explore which combinations of community-led and formal structures yield the most sustainable outcomes.

In summary, Community-For-Care proposes a pragmatic and compassionate reorientation of support: from a system centered primarily on patients and active caregivers to a broader ecology of care that continues beyond the end of caregiving. By recognizing post-caregivers as a group deserving structured attention and by distributing responsibility across the “village”, communities can foster more humane, responsive, and resilient environments for those who have dedicated part of their lives to caring for others.

## Figures and Tables

**Figure 1 healthcare-13-03318-f001:**
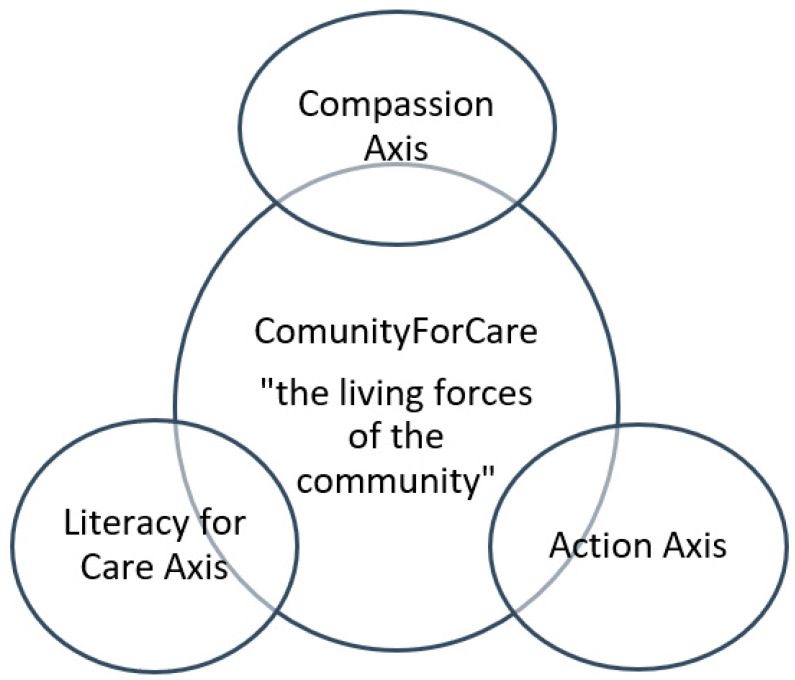
Thematic map of the proposed Community-For-Care Informal Post-Caregiver Support Network.

## Data Availability

The data presented in this study are available on request from the corresponding author (due to ethical and privacy restrictions, as they contain sensitive personal narratives and experiences shared by participants).

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
