# Peer review of "Community-for-Care: An Integrated Response to Informal Post-Caregivers"

_healthcare, 2025, doi:10.3390/healthcare13243318_

Round 1
Reviewer 1 Report
Comments and Suggestions for Authors
This study examines the conceptual framework of a community-centered support model for informal post-caregivers using a qualitative method. It presents an original approach titled "Community-For-Care: An Integrated Response to Informal Post-Caregivers." While the study is valuable for its focus on a significant social and health issue and its use of a multi-stakeholder data collection design, it needs improvement in terms of methodological clarity, theoretical depth, data presentation, and structural integrity. The following criticisms are structured to support the strengths of the text and enhance its scientific contribution.
- Lines 22–30: Although the Community-For-Care concept appears original, it is unclear how it differs from the Compassionate Communities model in the literature. The novelty of the approach should be more clearly stated.
- Lines 120–131: Although the study objectives are defined, the lack of clear and delimited research questions weakens the alignment with the methodology and findings.
- Lines 189–199: While conducting 3 focus groups with 15 participants is acceptable in qualitative studies, it is not explained why this number is sufficient for saturation. A justification for this should be provided.
- Lines 174–187: The fact that most participants are affiliated with care associations may lead to a bias towards an activist perspective in the results. This limitation should be emphasized.
- Lines 257–281: Although the coding process is explained in general terms, providing coding examples, criteria for theme development, and inter-rater reliability measures (Cohen's kappa) would elevate the study to a more scientific level.
- I think the explanation of Figure 1 is insufficient. It is stated that the figure conveys conceptual relationships, but the interaction between the axes is not explained in more technical terms; this makes it difficult for the reader to understand the diagram.
- There is a repetition of statements from the discussion section in the conclusions section. Furthermore, while the suggestions are broad, they could be made more concrete with actionable steps (an outline of a pilot program).
Overall, the study is valuable in that it addresses a critical societal need and employs a multi-stakeholder qualitative design. However, there is a need to clarify the research questions, provide a more detailed report of the methodological process, sharpen the positioning of the innovation claim within the existing literature, and present the findings in a more compact and less repetitive structure. For these reasons, it is premature to publish the article in its current form. My decision is for a major revision. If the authors address the improvements in the methodology, discussion, and conclusion sections, the study will be a much stronger candidate for publication.
Author Response
We sincerely thank the reviewer for the thorough and constructive feedback. We have revised the manuscript extensively to address all points raised. Below, we respond point-by-point, indicating changes made in the revised version (tracked in the manuscript).
- Lines 22–30 – Clarification of the novelty of Community-For-Care relative to Compassionate Communities
Reviewer’s comment: Although the Community-For-Care concept appears original, its distinction from the Compassionate Communities model is unclear.
Response:
We appreciate this observation. The revised manuscript now explicitly delineates how Community-For-Care extends and differentiates itself from Compassionate Communities, particularly through its exclusive focus on post-caregiving transitions and their specific identity, social, and health challenges. This clarification has been added in the revised Results and Conclusions sections.
- Lines 120–131 – Lack of explicit research questions
Reviewer’s comment: The study objectives are defined but not translated into clear research questions.
Response:
We agree. We have reformulated this section to include explicit, delimited research questions that align directly with the methodology and thematic findings. These are now included at the end of the Introduction.
- Lines 189–199 – Justification for sample size and saturation
Reviewer’s comment: The manuscript does not explain why three focus groups with 15 participants were sufficient for saturation.
Response:
A justification for sample adequacy has been added, referencing qualitative methodological literature on saturation in focus group research and explaining how thematic convergence was achieved across groups. This rationale has been introduced in the Participants subsection.
- Lines 174–187 – Potential bias due to strong representation of association members
Reviewer’s comment: The predominance of individuals affiliated with caregiver associations may bias results toward an activist perspective.
Response:
We acknowledge this important limitation. The revised manuscript now includes an explicit discussion of this potential bias and clarifies how it was considered during analysis. This is addressed both in the Participants section and in the Limitations.
- Lines 257–281 – Need for coding examples, criteria for theme development, and inter-rater reliability (Cohen’s kappa)
Reviewer’s comment: The coding process lacks detail; examples and reliability measures should be provided.
Response:
Thank you for this suggestion. We have strengthened the Data Analysis section in three concrete ways: Coding examples and criteria for theme development. We added a brief illustrative example of how a raw segment of text (e.g., reference to a “net with holes”) was initially coded and then grouped into a sub-theme and, ultimately, into the Compassion or Coordinated Action axis. We clarified the criteria used to develop themes: recurrence across groups, conceptual coherence, and explanatory power for the phenomenon of post-caregiver support.
Inter-rater reliability and Cohen’s kappa. Our analysis follows an interpretive, reflexive thematic analysis approach (Braun & Clarke), in which consensus-building and reflexive dialogue are emphasized rather than statistical inter-rater reliability. Nevertheless, we understand the reviewer’s concern and have now clarified our approach: we did not compute Cohen’s kappa, because our goal was not to produce highly standardized coding for quantitative generalization, but to develop a rich, conceptually grounded framework through investigator triangulation and consensus. We explicitly state this in the Methods and emphasise how rigour was ensured: independent coding, iterative comparison, discussion of discrepancies, and refinement of the codebook. For example: “In line with reflexive thematic analysis, we did not calculate a statistical inter-rater reliability index (e.g., Cohen’s kappa). Instead, rigour was ensured through investigator triangulation: both researchers independently coded the first transcript, compared and discussed their coding, and resolved discrepancies through consensus, refining the codebook iteratively. This process was repeated for subsequent transcripts to enhance credibility and dependability.”
- Figure 1 – Insufficient explanation of the conceptual relationships
Reviewer’s comment: The interaction between the axes is not adequately explained.
Response:
The explanation of Figure 1 has been substantially expanded to describe the technical logic of the model, the directionality of interactions, and the dynamic reinforcement between axes. The revised text offers a clearer interpretation of the conceptual framework.
- Discussion and Conclusions – Repetition and insufficient actionable recommendations
Reviewer’s comment: There is redundancy between discussion and conclusions, and recommendations could be more concrete.
Response:
The Conclusions section has been rewritten to avoid duplication and now includes more actionable elements, including an outline of a potential pilot program that operationalizes the Community-For-Care model at the municipal level.
General structural improvements requested
Reviewer’s comment: The study requires clearer research questions, more detailed methodology, stronger theoretical positioning, and a more concise presentation of findings.
Response:
These broader suggestions guided our revisions throughout the manuscript. We clarified the theoretical contribution, strengthened methodological descriptions, reorganized content to reduce repetition, and improved the coherence between aims, methods, results, and conclusions. All requested structural modifications have been implemented in the revised version.
We thank the reviewer for recommending a major revision and believe that the revised manuscript now addresses all methodological, conceptual, and structural issues raised. We are confident that the improvements significantly enhance the scientific contribution and clarity of the study.
Kind regards,
Catarina Afonso
Ana Spínola
Alcinda Reis
Susana Magalhães

Reviewer 2 Report
Comments and Suggestions for Authors
Thank you for the opportunity to review your manuscript. The study addresses an important and understudied topic: the needs, experiences, and support systems of informal post-caregivers. The qualitative design, the use of Narrative Medicine, the inclusion of multiple stakeholder perspectives, and the development of a preliminary conceptual model are notable strengths. However, several aspects would benefit from clarification or refinement to strengthen the manuscript’s conceptual coherence, methodological transparency, and overall scientific contribution.
Below are comments intended to improve the quality and readability of the study:
-
Introduction: The introduction provides a broad overview of the context and literature but becomes repetitive at times. Streamlining redundant information and more clearly articulating the specific knowledge gap addressed would strengthen the narrative. Clarifying how your work expands existing frameworks—such as Compassionate Communities and Narrative Medicine—would also improve coherence.
-
Methods: The integration of Narrative Medicine is innovative, yet it would be helpful to explain more concretely how this lens complemented the thematic analysis. The rationale for organizing focus groups by stakeholder type should be described in greater detail, noting both advantages and limitations. Additionally, outlining how coder disagreements were resolved would increase methodological transparency.
-
Results: The thematic findings are well supported, but some participant quotations are overly long or placed within dense paragraphs, which affects readability. Visually distinguishing participant quotes (e.g., through block quotes, indentation, or consistent formatting) would greatly enhance clarity. It may also be useful to indicate which findings were shared across groups and which were specific to certain stakeholders.
-
Discussion: The discussion engages well with the literature, though some points repeat results. A more concise synthesis of the study’s key contributions—especially regarding how the Community-For-Care model advances current thinking—would strengthen this section. Addressing practical challenges to implementation (e.g., institutional constraints, resource limitations) would also improve applicability.
-
Limitations: The limitations section could be expanded by noting that participants may be more motivated or sensitized than typical post-caregivers, by acknowledging the predominantly rural and female sample composition, and by considering how varying timeframes since caregiving ended may affect the narratives.
-
Conclusions: The conclusions are appropriate but would benefit from greater conciseness. Providing clearer suggestions for how this model could be piloted or operationalized in real settings would increase the practical impact of the study.
Author Response
We sincerely thank the reviewer for the thorough and constructive feedback. We have revised the manuscript extensively to address all points raised. Below, we respond point-by-point, indicating changes made in the revised version (tracked in the manuscript).
- Introduction – repetition, knowledge gap, link to frameworks
Reviewer comment: “The introduction provides a broad overview of the context and literature but becomes repetitive at times. Streamlining redundant information and more clearly articulating the specific knowledge gap addressed would strengthen the narrative. Clarifying how your work expands existing frameworks—such as Compassionate Communities and Narrative Medicine—would also improve coherence.”
Response: We agree that the introduction contained repetitions and that the knowledge gap required clearer articulation. We streamlined the section, clarified how the study expands upon Compassionate Communities and Narrative Medicine, and improved overall coherence.
Revisions have been incorporated in the Introduction.
- Methods – Narrative Medicine, FG rationale, coder disagreements
Reviewer comment: “The integration of Narrative Medicine is innovative, yet it would be helpful to explain more concretely how this lens complemented the thematic analysis. The rationale for organizing focus groups by stakeholder type should be described in greater detail, noting both advantages and limitations. Additionally, outlining how coder disagreements were resolved would increase methodological transparency.”
Response: We agree that methodological clarifications were needed. The integration of Narrative Medicine, the rationale for focus group composition, and the procedure for resolving coder disagreements have all been explained in more detail.
Revisions have been added in the Methods section.
- Results – quotations, readability, cross-group vs specific findings
Reviewer comment: “The thematic findings are well supported, but some participant quotations are overly long or placed within dense paragraphs, which affects readability. Visually distinguishing participant quotes (e.g., through block quotes, indentation, or consistent formatting) would greatly enhance clarity. It may also be useful to indicate which findings were shared across groups and which were specific to certain stakeholders.”
Response: We agree with the recommendation to improve readability by reformatting long quotations. Participant quotes are now visually distinguished, and we clarify which findings were shared across all groups and which emerged from specific stakeholders.
Revisions have been made in the Results section.
- Discussion – repetition, key contributions, practical challenges
Reviewer comment: “The discussion engages well with the literature, though some points repeat results. A more concise synthesis of the study’s key contributions—especially regarding how the Community-For-Care model advances current thinking—would strengthen this section. Addressing practical challenges to implementation (e.g., institutional constraints, resource limitations) would also improve applicability.”
Response: We agree that some redundancy with the Results section needed reduction. The Discussion now offers a more concise synthesis of contributions and includes a clearer explanation of how the Community-For-Care model advances current knowledge, as well as practical considerations for implementation. Revisions have been made in the Discussion section.
- Limitations – sample characteristics and sensitization
Reviewer comment: “The limitations section could be expanded by noting that participants may be more motivated or sensitized than typical post-caregivers, by acknowledging the predominantly rural and female sample composition, and by considering how varying timeframes since caregiving ended may affect the narratives.”
Response: We agree with the reviewer’s suggestions. The limitations section was expanded to address possible motivation/sensitization bias among participants, the predominantly rural and female sample, and the influence of different post-care timeframes.
Revisions have been incorporated in the Limitations section.
- Conclusions – conciseness and operationalization
Reviewer comment: “The conclusions are appropriate but would benefit from greater conciseness. Providing clearer suggestions for how this model could be piloted or operationalized in real settings would increase the practical impact of the study.”
Response: We agree that greater conciseness and more actionable suggestions were necessary. The Conclusions have been streamlined and now include clearer possibilities for piloting and operationalizing the Community-For-Care model. Revisions have been made in the Conclusions section.
All suggested changes have been integrated into the revised manuscript.
We thank the reviewer for recommending a major revision and believe that the revised manuscript now addresses all methodological, conceptual, and structural issues raised. We are confident that the improvements significantly enhance the scientific contribution and clarity of the study.
Kind regards,
Catarina Afonso
Ana Spínola
Alcinda Reis
Susana Magalhães

Round 2
Reviewer 1 Report
Comments and Suggestions for Authors
Accept in present form
Reviewer 2 Report
Comments and Suggestions for Authors
The changes made by the authors adequately and satisfactorily address the requirements raised by this reviewer.